# Electrical Impedance Tomography of Industrial Two-Phase Flow Based on Radial Basis Function Neural Network Optimized by the Artificial Bee Colony Algorithm

**DOI:** 10.3390/s23177645

**Published:** 2023-09-04

**Authors:** Zhiheng Zhu, Gang Li, Mingzhang Luo, Peng Zhang, Zhengyang Gao

**Affiliations:** 1School of Electronic Information and Electrical Engineering, Yangtze University, Jingzhou 434023, China; 2021710529@yangtzeu.edu.cn (Z.Z.); lmz@yangtzeu.edu.cn (M.L.); 201700997@yangtzeu.edu.cn (P.Z.); 2021710538@yangtzeu.cn (Z.G.); 2Key Laboratory of Bridge Engineering Safety Control by Department of Education, Changsha University of Science and Technology, Changsha 410076, China

**Keywords:** electrical impedance tomography (EIT), artificial bee colony (ABC), radial basis function neural network (RBFNN), image reconstruction, two-phase flow

## Abstract

In electrical impedance tomography (EIT) detection of industrial two-phase flows, the Gauss-Newton algorithm is often used for imaging. In complex cases with multiple bubbles, this method has poor imaging accuracy. To address this issue, a new algorithm called the artificial bee colony–optimized radial basis function neural network (ABC-RBFNN) is applied to industrial two-phase flow EIT for the first time. This algorithm aims to enhance the accuracy of image reconstruction in electrical impedance tomography (EIT) technology. The EIDORS-v3.10 software platform is utilized to generate electrode data for a 16-electrode EIT system with varying numbers of bubbles. This generated data is then employed as training data to effectively train the ABC-RBFNN model. The reconstructed electrical impedance image produced from this process is evaluated using the image correlation coefficient (ICC) and root mean square error (RMSE) criteria. Tests conducted on both noisy and noiseless test set data demonstrate that the ABC-RBFNN algorithm achieves a higher ICC value and a lower RMSE value compared to the Gauss–Newton algorithm and the radial basis function neural network (RBFNN) algorithm. These results validate that the ABC-RBFNN algorithm exhibits superior noise immunity. Tests conducted on bubble models of various sizes and quantities, as well as circular bubble models, demonstrate the ABC-RBFNN algorithm’s capability to accurately determine the size and shape of bubbles. This outcome confirms the algorithm’s generalization ability. Moreover, when experimental data collected from a 16-electrode EIT experimental device is employed as test data, the ABC-RBFNN algorithm consistently and accurately identifies the size and position of the target. This achievement establishes a solid foundation for the practical application of the algorithm.

## 1. Introduction

Electrical impedance tomography (EIT) is a non-invasive imaging technique that uses boundary voltage measurements to reconstruct the distribution of electrical conductivity within a domain. Since its inception in the 1980s, EIT has adopted widespread applications in biomedicine [1], damage detection [2], and industrial two-phase flow detection [3]. The detection of gas–liquid two-phase flow in pipes holds significant importance in industrial activities such as chemical production, oil and gas storage and transportation, thermal dynamics, and aerospace engineering.

EIT, as a measurement technique, has gained momentum alongside other imaging techniques such as the gamma-ray attenuation technique [4], X-ray attenuation technique [5], and acoustic imaging [6]. In terms of affordability, non-invasiveness, and ease of implementation [7], EIT holds advantages over other methods.

The Gauss–Newton algorithm is commonly used to perform imaging of the gas–liquid two-phase flow in pipelines using EIT. However, this algorithm falls short of accuracy and precision when faced with complex bubble shapes or an abundance of bubbles in the domain. In recent years, the development of deep learning in image processing has provided a new direction for the advancement of EIT technology. Neural networks can achieve non-linear mapping from input boundary voltage values to output conductivity values, thereby reducing the ill-posed nature of traditional image reconstruction methods and improving the imaging precision of electrical impedance tomography [8].

Researchers have utilized various deep learning techniques to improve EIT imaging capabilities [9]. The generative adversarial network algorithm has been employed by Chen et al. to enhance the detailed features of the imaging results; however, the resulting image still exhibits a relatively low resolution [10]. Hamilton et al. utilized the U-net algorithm to further fit the EIT images obtained using the D-bar algorithm to develop a real-time EIT algorithm that is applicable to various datasets [11]. Li et al. implemented a fully connected neural network for multi-object imaging, but it exhibited limited generalization ability [12]. Guilherme C. Duran et al. developed a convolutional neural network-based method for EIT image reconstruction that enables imaging of irregular objects [13]. Nevertheless, this method’s noise resistance is limited, and it generates a notable number of artifacts. Considering the potential need for more sophisticated imaging requirements in industrial two-phase flow detection, enhancing the anti-noise and generalization capabilities of imaging algorithms becomes imperative.

To address the aforementioned challenges of the poor anti-noise and generalization capabilities of existing algorithms in gas–liquid two-phase flow detection using EIT, we have introduced a novel image reconstruction algorithm, the artificial bee colony–radial basis function neural network (ABC-RBFNN), into EIT. The radial basis function neural network (RBFNN) is a feedforward neural network that possesses a simple network structure and strong non-linear approximation capabilities [14]. However, the control parameters of RBFNN have a significant impact on its output, leading to the generation of numerous artifacts in EIT image reconstruction. However, determining the optimal parameters of neural networks remains challenging. On the other hand, bio-inspired optimization algorithms exhibit strong optimization capabilities. Among the numerous bio-inspired algorithms, the artificial bee colony algorithm not only has fewer control parameters, but also demonstrates robust global optimization capabilities. Khurram Hussain et al. used a hybrid of the WOA-ABC and proposed a CNN for intrusion detection systems in wireless sensor networks [15]. Ebrahim Noroozi-Ghaleini et al. have also employed the artificial bee colony algorithm to optimize feedforward neural networks, thereby enhancing the accuracy of predictions [16]. In order to determine the optimal control parameters for RBFNN and mitigate the occurrence of excessive artifacts, we employed the artificial bee colony algorithm to optimize the three parameters of RBFNN, namely weight, center vector, and basis radius, to remove artifacts.

To evaluate the performance of ABC-RBFNN, we simulated a 16-electrode EIT system using the EIDORS software platform forward program to create 15,000 sets of electrode potential simulation samples. We allocated 90% of the samples to the training set to train the network model and used the remaining 10% as the testing set to verify the image reconstruction performance of the algorithm. We compared the performance of the ABC-RBFNN with the Gauss–Newton algorithm and RBFNN algorithm. Furthermore, we performed tests to evaluate the anti-noise and generalization capabilities of the ABC-RBFNN. Finally, we validated the feasibility of our proposed algorithm for practical applications by conducting experiments using a 16-electrode EIT system to collect real data.

## 2. The Principles Related to EIT Technology

### 2.1. The Principles of the EIT Technology Measurement System

The measurement system for EIT technology mainly consists of three parts: current excitation, data acquisition, and display imaging. The 16-electrode EIT system principle is illustrated in Figure 1. Sixteen copper electrodes are evenly arranged on the wall of the device. This paper uses the ad–ad mode, which is the adjacent electrode excitation–adjacent electrode acquisition mode. The process involves selecting a pair of adjacent electrodes as the excitation electrodes, injecting an excitation current of constant amplitude and frequency and then selecting another pair of electrodes as the measurement electrodes to collect the voltage data. This process is repeated until all remaining adjacent electrode pairs are utilized as measurement electrodes. Subsequently, the above process is repeated until all the adjacent electrode pairs have been used as excitation electrodes.

### 2.2. Mathematical Model for EIT Image Reconstruction

The EIT measurement field satisfies Maxwell’s equations and electromagnetic field theory [17]. The mathematical model for EIT image reconstruction is given as follows:(1)∇⋅(σ(x,y)∇Φ(x,y))=0,(x,y)∈Ω

In Equation (1), *σ*(*x*,*y*) represents the distribution of electrical conductivity within the field, *Φ*(*x*,*y*) represents the distribution function of the electric potential in the field, and *Ω* represents the field domain.

The boundary conditions for the EIT field are set as follows:(2)$$\frac{\sigma (x,y)\partial \Phi (x,y)}{\partial n}=-j(x,y),(x,y)\in \partial \Omega$$
(3)Φ(x,y)=U(x,y)

In Equations (2) and (3), ***n*** represents the outward normal unit vector of the boundary of the field; *j*(*x*,*y*) represents the current density of the injected current on the boundary, where the current density is 0 in the absence of current injection; *∂Ω* represents the boundary of the field; and *U*(*x*,*y*) represents the potential distribution on the boundary of the field.

If a constant current source is used as the excitation current in the experiment, i.e., the frequency and amplitude of the excitation current are constant, then the EIT image reconstruction becomes the exploration of the relationship between the conductivity distribution *σ* and the potential distribution *Φ* inside the field. The process of solving for the potential distribution *Φ* with a known conductivity distribution *σ* in the field is called the forward problem of EIT, while the inverse problem of EIT refers to the process of solving for the conductivity distribution σ with a known potential distribution *Φ*.

The inverse problem of EIT is a nonlinear imaging issue. Compared to other neural networks, RBFNN’s advantage lies in its strong global approximation capability for nonlinear models. However, determining the optimal control parameters of RBFNN is challenging, often necessitating the combined use of bio-inspired optimization algorithms.

EIT image reconstruction is the solution of the inverse problem of EIT. Due to the limited control parameters and strong optimization capability of the artificial bee colony algorithm, in this paper, a method based on the artificial bee colony algorithm and optimized radial basis function neural network is proposed to solve the inverse problem of EIT. The voltage values of the EIT system’s measuring electrodes are used as the inputs of the network, and the conductivity inside the field is the output of the network.

## 3. The Artificial Bee Colony Algorithm–Optimized Radial Basis Function Neural Network

### 3.1. The Principle of Radial Basis Function Neural Networks

The radial basis function neural network (RBFNN) is a classic three-layer feedforward neural network. Its basic structure consists of an input layer, a hidden layer, and an output layer, as shown in Figure 2. Due to the non-linear transformation from the output layer to the hidden layer and the linear transformation from the hidden layer to the output layer, the RBFNN is capable of approximating any non-linear function and has advantages such as fast learning convergence and a good generalization ability [18].
(4)$$h_{m}(X)=e^{\left(-\frac{∥X-o_{m}{∥}^{2}}{2b_{m}^{2}}\right)}$$

In the network structure, the input layer is ***X*** = [*x_1_*, …, *x_n_*]*^T^*, and the hidden layer is the Gaussian function ***H*** = [*h_1_*, …, *h_m_*]*^T^*.

In Equation (4), ***B*** = [*b_1_*, …, *b_m_*]*^T^* is the basis width vector, ***O*** = [*o_1_*, …, *o_m_*]*^T^* is the center vector, and the weight vector from the hidden layer to the output layer is ***W*** = [*w_1_*, …, *w_i_*]*^T^*. The predicted output ***Y*** = [*y_1_*, …, *y_i_*]*^T^* of the RBFNN is calculated using Equation (5).
(5)Y=H⋅W

During training, there are three unknown parameters to be learned: center vectors ***O***, width vectors ***B***, and weight vectors ***W***. Since these three parameters directly affect the reliability of the radial basis function neural network, in this study, two different training methods were employed to optimize these three parameters. The first method involved using the RBFNN implementation within the MATLAB Neural Network Toolbox. The second method treated these three parameters as an optimization problem, utilizing the ABC algorithm to determine their optimal values, thus resulting in the ABC-RBFNN approach. The initial parameter setting for the RBFNN, specifically the number of neurons in the hidden layer, was consistently set to 1500 neurons.

### 3.2. The Principle of the Artificial Bee Colony Algorithm

The artificial bee colony (ABC) algorithm is an intelligent optimization algorithm proposed by simulating the process of honeybee foraging [19]. In the ABC algorithm, the solution represents the location of a nectar source, and the fitness value represents the amount of nectar collected at that nectar source. The process of searching for better nectar sources is essentially the process of searching for higher-quality solutions. The algorithm divides all bees into three types of roles: (a) employed bees, responsible for searching for new nectar sources and sharing information about the nectar source with onlooker bees; (b) onlooker bees, who choose their own search targets and directions based on the information provided by employed bees and continue to search for new nectar sources; and (c) scout bees, who adopt this role when the number of searches near a nectar source reaches a set upper limit. Scout bees search for better nectar sources to avoid getting trapped in local optima. The algorithm consists of several phases, which are described below.

#### 3.2.1. The Initialization Phase of the Bee Colony

The initialization stage of the algorithm involves setting up the algorithm parameters, including the total number of nectar sources *n* (Set to 100), the search limit per nectar source limit (Set to 50), and the termination condition N (Set to 250). In the artificial bee colony algorithm, the total number of nectar sources is equal to the number of employed bees, which is also equal to the number of onlooker bees. The initial nectar sources are generated using the method shown in Equation (6):(6)$$x_{ij}=x_{j}^{\min}+rand[0,1]\left(x_{j}^{\max}-x_{j}^{\min}\right)$$

In Equation (6), *x_ij_* represents the *j*-th dimension of the *i*-th nectar source *x_i_*, where *i* = 1, 2, 3, …, *n*; *j* = 1, 2, 3, …, *m*; and *m* depends on the dimension of the optimization problem. *x_j_^min^* and *x_j_^max^* represent the minimum and maximum values of the *j*-th dimension space, respectively. *rand*[0, 1] represents a random number in the range of [0, 1]. *n* initial nectar sources are generated randomly using this equation.

#### 3.2.2. Employed Bee Phase

During the employed bee phase, new nectar sources are searched for using Equation (7):(7)$$v_{ij}=x_{ij}+rand[-1,1]\left(x_{ij}-x_{kj}\right)$$

In Equation (7), *x_ij_* and *x_kj_* denote the *j*-th dimension of the *i*-th nectar source *x_i_* and the *k*-th nectar source *x_k_* in the neighborhood *k* = 1, 2, 3, …, *n* (*k* ≠ *i*), respectively. *rand*[−1, 1] indicates a random number within the range of [−1, 1]. *v_ij_* represents the updated *j*-th dimension of the nectar source.

The employed bees calculate the fitness of all the nectar sources after finding them. The fitness value reflects the quality of the nectar source, with a larger fitness value indicating a better-quality nectar source. The employed bees keep the high-quality nectar sources and share this information with the onlooker bees.

#### 3.2.3. Onlooker Bee Phase

The onlooker bee phase starts after the completion of the employed bee phase. Based on the information provided by the employed bees and using Equation (8), the probability is calculated to select high-quality nectar sources for exploitation, ensuring a higher probability of exploiting nectar sources with higher fitness. The method for updating the nectar sources is the same as in the employed bee phase using Equation (7), with high-quality nectar sources being retained.
(8)$$P_{i}=\frac{F\left(x_{i}\right)}{\sum_{i=1}^{n}F\left(x_{i}\right)}$$

In Equation (8), *P_i_* represents the probability of exploitation, F(*x_i_*) represents the fitness of the *i*-th nectar source *x_i_*, and *n* represents the total number of nectar sources.

#### 3.2.4. Scout Bee Phase

If a nectar source is not updated after being exploited many times, and the number of such times exceeds the upper limit “limit” set, then the scout bees need to randomly generate new nectar sources according to Equation (6).

### 3.3. Radial Basis Function Neural Network Optimized Using the Artificial Bee Colony Algorithm

Using the artificial bee colony algorithm to optimize the radial basis function neural network involves the following specific optimization process:The first step in optimizing the radial basis function neural network using the artificial bee colony algorithm is to initialize the parameters of the algorithm. The initial nectar sources are generated using Equation (6).During the employed bee phase, the employed bees search for new nectar sources using Equation (7), the fitness of all the nectar sources is evaluated using Equation (9), and the fitness values of the old nectar sources are compared with the fitness values of the newly generated nectar sources. The higher-quality nectar sources are retained, which are the ones with larger fitness values. The fitness function of the standard ABC algorithm is defined by Equation (9):
(9)$$F\left(x_{i}\right)=\left\{\begin{array}{c} \frac{1}{1+f_{i}},f_{i}⩾0\\ 1+abs\left(f_{i}\right),f_{i}<0 \end{array}\right.$$

In Equation (9), *f_i_* represents the function value of the solution, F(*x_i_*) represents the fitness of the *i*-th nectar source *x_i_*, and abs denotes the absolute value.

In the employed bee phase and the onlooker bee phase, the ABC algorithm essentially involves the employed bees and onlooker bees searching for solutions of higher quality, i.e., greater fitness values, to be used for updating, considering that a smaller RMSE of the RBFNN can lead to better imaging quality and fewer artifacts in EIT. In order to achieve a smaller RMSE for RBFNN, in this paper, the fitness function of the standard ABC algorithm is modified to use the reciprocal of the root mean square error (RMSE) of the radial basis function as the fitness function, which is defined in Equation (10):(10)$$F\left(x_{i}\right)=\frac{1}{\sqrt{\frac{1}{N}\sum_{i=1}^{N}{\left(Y_{i}-P_{i}\right)}^{2}}}$$

In Equation (10), ***Y****_i_* represents the predicted conductivity value, ***R****_i_* represents the true conductivity value, *N* represents the sample size, and F(*x_i_*) represents the fitness of the *i*-th nectar source *x_i_*.

Compared to the reciprocal of RMSE, the employed bee phase and the onlooker bee phase of the ABC algorithm contribute less to the improvement of imaging quality in industrial two-phase flow EIT imaging when searching for conventional fitness functions. Simultaneously, when combining the simple network structure and fast learning of RBFNN with the low parameter count of the ABC algorithm, the computational burden of the ABC-RBFNN algorithm remains manageable. This approach also offers the advantages of a short training time and fewer control parameters.

3.The onlooker bees calculate the extraction probability of the nectar sources based on the information provided by the employed bees using Equation (8) and select the nectar sources with a high probability. Then, they search for new nectar sources near the chosen nectar sources using Equation (7) and calculate the fitness of the nectar sources using Equation (10). Lastly, they record higher-quality nectar sources.4.If the search at a certain nectar source reaches the set upper limit of the search times, the employed bee is transformed into a scout bee to generate a new nectar source.5.Steps 2 to 4 are repeated until the termination criteria of the algorithm are met, and the global optimum solution is outputted.6.The global optimal solution is incorporated into the parameters of the radial basis function neural network.

The flowchart of this process is shown in Figure 3.

## 4. Simulation and Results

### 4.1. Dataset

The training process of a neural network requires a large amount of data, and the generalization ability of the network is greatly influenced by the quality of the dataset. However, in reality, it is extremely difficult to obtain a large number of data samples for EIT systems. Therefore, a simulation-based EIT system using the EIDORS software platform forward program was used to generate the dataset. The circular physical field was divided into 576 elements in the simulation program.

The simulation system was a 16-electrode EIT system which adopts the ad–ad mode, i.e., adjacent current excitation and adjacent voltage acquisition mode. The diameter of the circular pipe was 0.2 m (conductivity is 10^−3^ S/m). One, two, and three circular models were generated in the simulation to simulate bubbles, and the diameter of each circular model was 0.02 m (conductivity is 0 S/m).

For each case, 5000 data samples were randomly generated for 1, 2, and 3 circular objects in the circular measurement area, respectively, resulting in a total of 15,000 data samples. Figure 4 represents a schematic of the conductivity distribution within a partial dataset. A total of 4500 data samples were completely randomly taken as the training set for each case, and the remaining 500 samples were used as the test set, resulting in 10,500 data samples for training and 1500 for testing. The training set and the test set did not overlap.

Each datapoint in the dataset included the conductivity data of 576 elements and the corresponding 208 measured voltage data. The collected voltage data and corresponding conductivity data were normalized to the interval [0,1] using Equations (11) and (12), respectively.
(11)$$U_{i}=\frac{U_{i}-U_{\min}}{U_{\max}-U_{\min}}$$
(12)$$\sigma_{i}=\frac{\sigma_{i}-\sigma_{\min}}{\sigma_{\max}-\sigma_{\min}}$$

### 4.2. Evaluation Criteria for the Algorithm

To evaluate the image reconstruction quality of the different algorithms for the EIT system, the root mean square error (RMSE) and the image correlation coefficient (ICC) were used as the evaluation criteria. The calculation methods for the RMSE and ICC are represented by Formulas (13) and (14), respectively.
(13)$$RMSE=\sqrt{\frac{1}{n}\overset{n}{\underset{i=1}{\sum }}{\left(R_{i}-Y_{i}\right)}^{2}}$$
(14)$$ICC=\frac{\sum_{i=1}^{n}\left(Y_{i}-\bar{Y}\right)\left(R_{i}-\bar{R}\right)}{\sqrt{\sum_{i=1}^{n}{\left(Y_{i}-\bar{Y}\right)}^{2}\sum_{i=1}^{n}{\left(R_{i}-\bar{R}\right)}^{2}}}$$

In Equations (13) and (14), ***Y****_i_* and $\bar{Y}$ represent the predicted and average conductivity values, respectively. ***R****_i_* and $\bar{R}$ represent the true and average conductivity values, respectively. *n* represents the number of divided elements.

### 4.3. Simulation Results of the Noise-Free Test Set

This paper conducted simulation experiments on image reconstructions using the Gauss–Newton algorithm, the built-in RBFNN algorithm in MATLAB, and the ABC-RBFNN algorithm proposed in this paper. A cross-comparison between the different algorithms was used to verify the effectiveness of the proposed algorithm for image reconstruction. The image reconstruction results of the three different algorithms for different target numbers in the noise-free test set are shown in Figure 5, and the corresponding average RMSE and ICC values are shown in Figure 6.

Based on the imaging results in Figure 5, it can be observed that compared to the Gauss–Newton algorithm, the ABC-RBFNN algorithm provides clear and accurate reconstructions of the size, shape, and number of targets. In comparison to the RBFNN algorithm, the ABC-RBFNN method results in fewer artifacts and clearer contours for cases with two or more targets. The results shown in Figure 6 demonstrate that the ABC-RBFNN algorithm proposed in this study had the lowest RMSE (RMSE = 0.0351) and the highest ICC (ICC = 0.9622), indicating that the proposed ABC-RBFNN algorithm outperforms the other two algorithms in terms of image reconstruction.

### 4.4. Noise Resistance Testing of the Algorithm

Different levels of Gaussian white noise (30 dB, 40 dB, and 50 dB) were added to the test set to evaluate the algorithm’s noise resistance through cross-comparisons. The image reconstruction results with added noise of different signal-to-noise ratios are shown in Figure 7, where the number of targets can still be distinguished even at a maximum noise level of 30 dB. Using the ABC-RBFNN algorithm, the average RMSE and ICC values were calculated for each level of added noise and are presented in Figure 8. The results demonstrate that the average ICC value remained above 0.87, and the average RMSE value remained below 0.1. These findings indicate that the ABC-RBFNN algorithm employed in this study effectively distinguishes the number of targets and accurately determines their positions with clarity. This outcome illustrates the algorithm’s robustness against noise interference.

### 4.5. Generalization Test of the Algorithm

In industrial two-phase flows, various patterns with different shapes and sizes may occur. Therefore, it is crucial for the ABC-RBFNN algorithm to handle targets with different sizes and annular flow patterns. To test the algorithm’s generalization capability, simulation models that were not included in the training set were adopted. A comparison was made between the Gauss–Newton, RBFNN, and ABC-RBFNN algorithms. Given that other shapes of flow patterns may appear and their sizes may not be exactly the same, the ABC-RBFNN algorithm must deal with different target sizes and annular flow patterns. Simulation models that did not appear in the training set or the test set were used to test the generalization ability of the proposed algorithm compared with the Gauss–Newton and RBFNN algorithms. The image reconstruction results of the annular models and models of different sizes are shown in Figure 9, while the corresponding average ICC and RMSE values are shown in Figure 10.

Based on the image reconstruction results, it can be seen that when dealing with shapes and sizes of objects that were not present in the training and testing sets, both the RBFNN and ABC-RBFNN were able to reconstruct the size and position of the objects. However, the objects reconstructed using the ABC-RBFNN algorithm exhibited clearer contours with fewer artifacts compared to the RBFNN. In Model 2, the Gauss–Newton algorithm generated a large number of artifacts, resulting in overlapping contours. Meanwhile, the RBFNN algorithm was not as good as the ABC-RBFNN algorithm at capturing details. In Model 3, the Gauss–Newton algorithm failed to identify a circular object, whereas the RBFNN algorithm produced a large number of artifacts near the circular object, resulting in unclear contour identification. In Model 4, the Gauss–Newton algorithm identified the ring-shaped object as a complete circular object, while the RBFNN algorithm produced a small number of artifacts near the ring-shaped object, resulting in a slightly thicker ring. The results shown in Figure 10 further support that the ABC-RBFNN algorithm has a better generalization ability compared to the other two algorithms. These results confirm the strong generalization ability of the ABC-RBFNN algorithm proposed in this study.

## 5. Experimental Validation

The experimental data were measured using a 16-electrode EIT system. The experimental setup, as shown in Figure 11, consisted of a measurement container, a data acquisition system, and an inversion imaging system. The measurement system consisted of a 200 mm diameter acrylic bucket and 16 copper electrode plates placed on the inner wall of the bucket. The bucket was filled with salt water (conductivity is 10^−3^ S/m) to simulate the liquid phase in the pipeline, and a solid rubber rod (conductivity approximated to be 0 S/m) with a diameter of 20 mm was placed in the bucket to simulate the gas phase in the pipeline. The experiment utilized the potential data collected from each electrode multiple times and used the ABC-RBFNN algorithm to invert the position and size of the rubber rod. The ABC-RBFNN algorithm was compared with the two other algorithms to verify its effectiveness.

Figure 12 displays partial experimental imaging results. The left image illustrates the solid rubber rod positioned at various positions and with different quantities within the EIT measurement container. The first column showcases the imaging outcomes of the Gauss–Newton algorithm during the experiment. Noticeable artifacts were apparent, and the size recognition of the solid rubber rod was inaccurate. However, the positions and quantities were correctly identified.

In the second column, the imaging results obtained using the RBFNN algorithm are presented. The RBFNN algorithm exhibited errors in recognizing the number and position of the rubber rod, leading to a multitude of noticeable artifacts. As a result, discerning the size, position, and quantity of the solid rubber rod was challenging.

In the third column, the imaging results of the ABC-RBFNN algorithm are presented. The red area in the image closely matches the size, position and quantity of the solid rubber rod depicted in the first column. There are minimal artifacts observed in the middle of the image in the third column.

In summary, the experimental results highlight that the ABC-RBFNN algorithm achieved more accurate information regarding the size and position of the solid rubber rod compared to the other two algorithms tested. Nevertheless, it is worth noting that the imaging quality of the measured data was inferior to that of the simulation data set due to the presence of both systematic and random errors in the collected data. These errors can be partially attributed to the production process of the measurement device, where variations in electrode sizes and deviations in their positions from the theoretical design introduce errors. Moreover, the presence of noise in the data acquisition circuit adds to the overall error. Figure 13 provides additional evidence supporting the aforementioned observations.

Despite the lower imaging quality of the measured data compared to the simulation data set, the experimental results successfully confirmed the ABC-RBFNN algorithm’s capability to accurately reconstruct the size and position of the solid rubber rod, even in the presence of noise interference. These findings highlight the algorithm’s robustness against noise and its feasibility for practical applications.

## 6. Conclusions

To enhance image reconstruction quality in electrical impedance tomography (EIT), this study introduces the ABC-RBFNN algorithm for the first time. The proposed algorithm employs the artificial bee colony optimization algorithm to determine the optimal network centers, network radial basis function widths, and connection weights, thus significantly enhancing the prediction accuracy of the network model. The model was trained using 15,000 EIT simulation datapoints generated using the EIDORS software platform forward program and was subsequently tested for reliability. The image reconstruction results obtained from the noise resistance testing and generalization testing validate the robust noise resistance and generalization ability of the ABC-RBFNN algorithm. This confirms its capability to handle complex scenarios effectively. Moreover, the experimental data collected using a 16-electrode EIT experimental device further demonstrate the algorithm’s ability to accurately identify the number and position of multiple target objects, even in the presence of real-world noise, resulting in clear imaging. This underscores the algorithm’s practical feasibility. Overall, in comparison to the Gauss–Newton algorithm and the RBFNN algorithm, the ABC-RBFNN algorithm proposed in this study exhibits superior prediction accuracy, greater noise resistance, and provides clearer image reconstruction results.

## Figures and Tables

**Figure 1 sensors-23-07645-f001:**
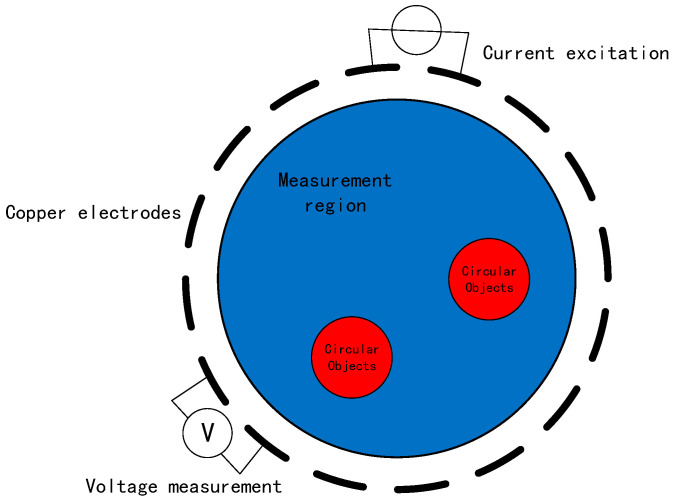
EIT measurement principle.

**Figure 2 sensors-23-07645-f002:**
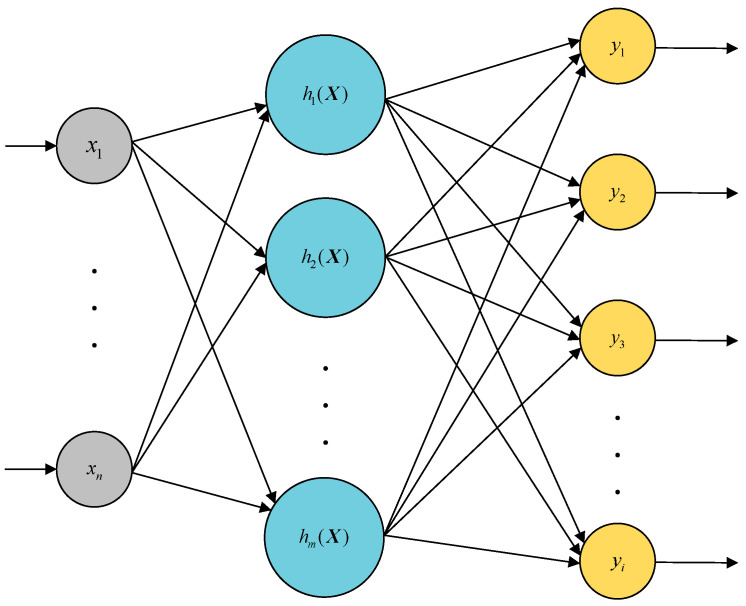
RBFNN structure.

**Figure 3 sensors-23-07645-f003:**
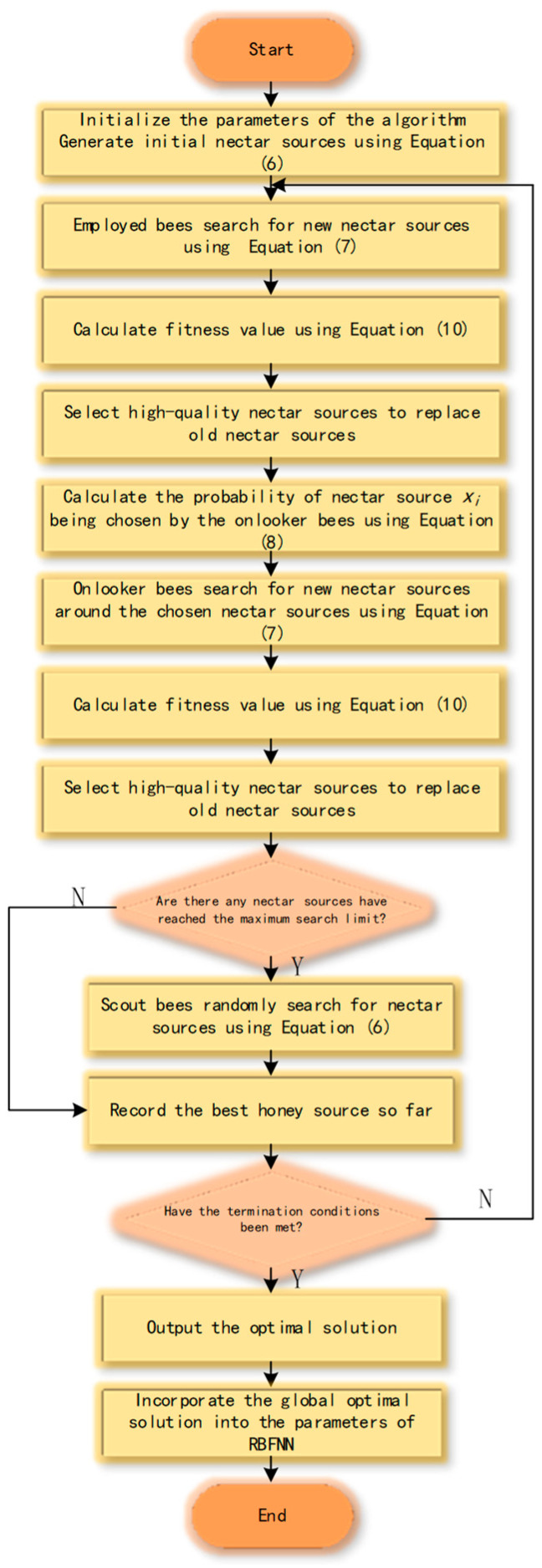
Flowchart of the ABC-RBFNN.

**Figure 4 sensors-23-07645-f004:**
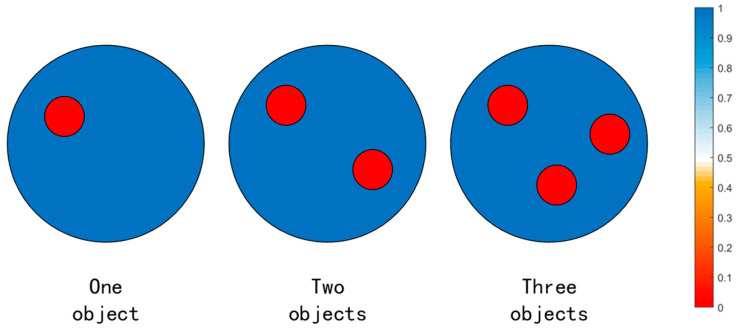
Schematic of the conductivity distribution within a partial dataset.

**Figure 5 sensors-23-07645-f005:**
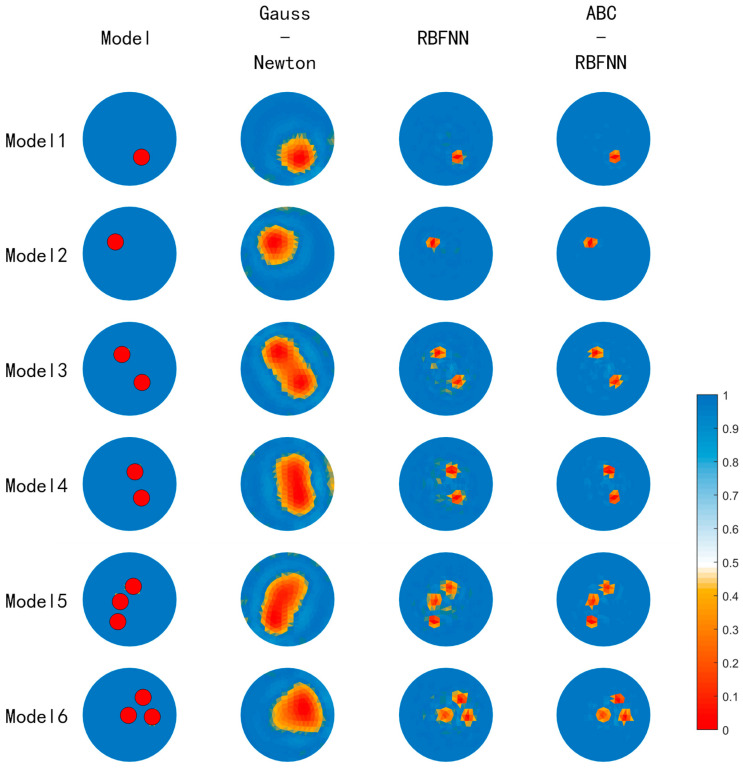
Image results of the test set without noise.

**Figure 6 sensors-23-07645-f006:**
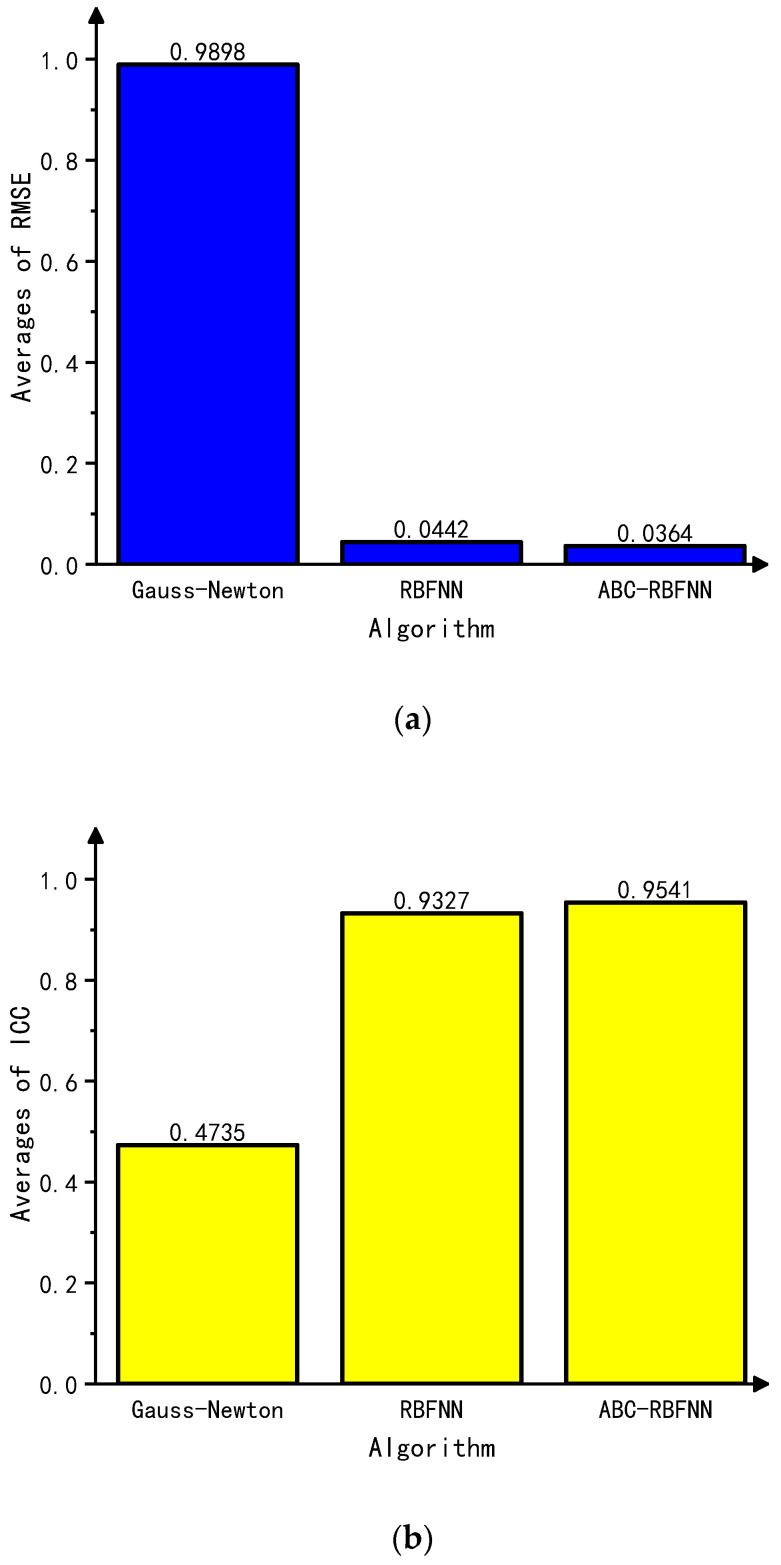
Average RMSE and ICC values of the test set without noise. (**a**) Average RMSE values of the test set without noise. (**b**) Average ICC values of the test set without noise.

**Figure 7 sensors-23-07645-f007:**
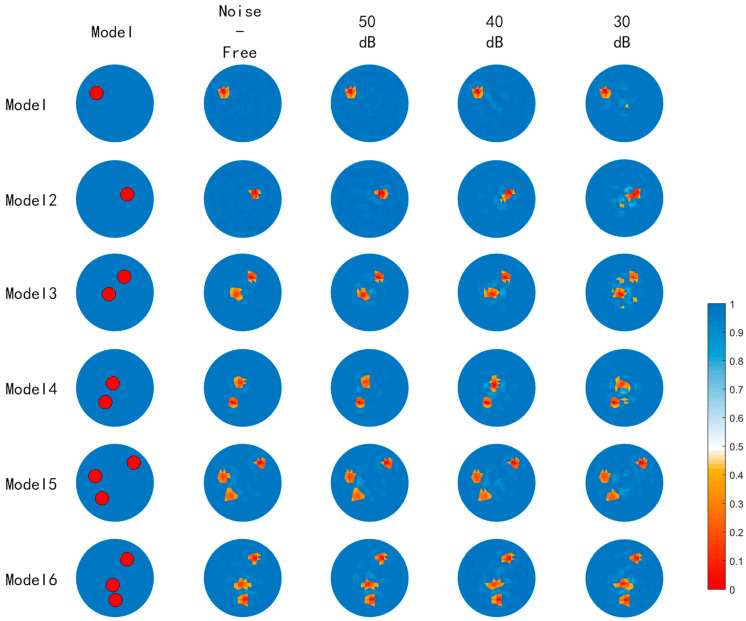
Image results of the test set with noise.

**Figure 8 sensors-23-07645-f008:**
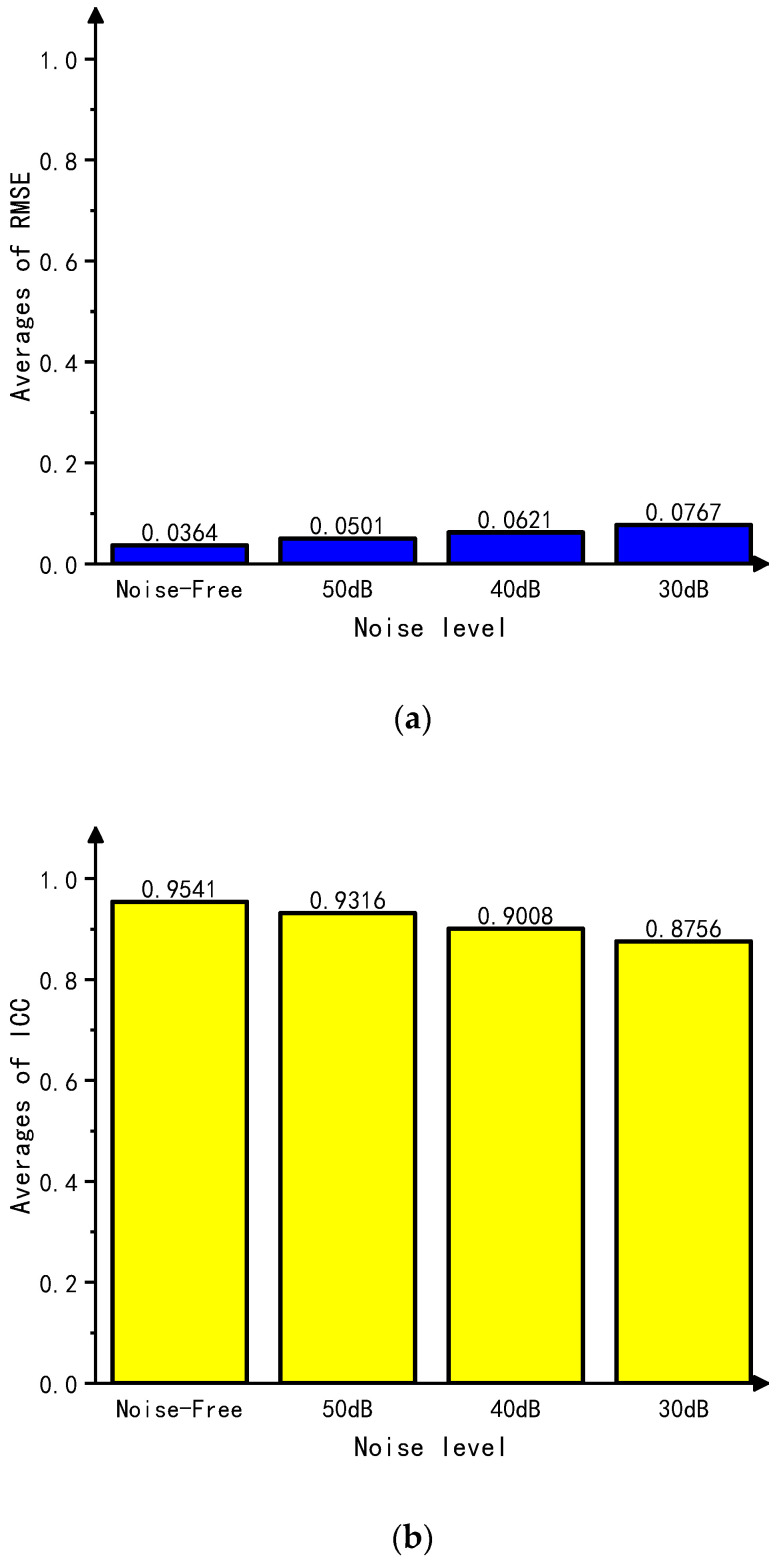
Average RMSE and ICC values of the test set with noise. (**a**) Average RMSE values of the test set with noise. (**b**) Average ICC values of the test set with noise.

**Figure 9 sensors-23-07645-f009:**
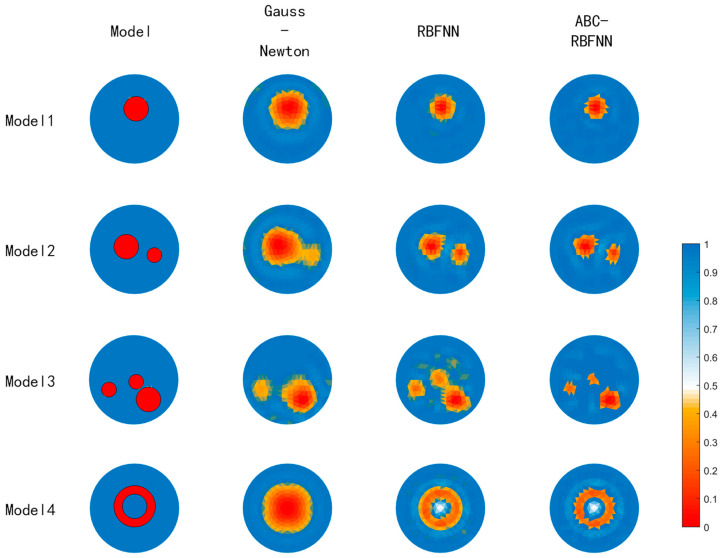
Image results of the generalization test.

**Figure 10 sensors-23-07645-f010:**
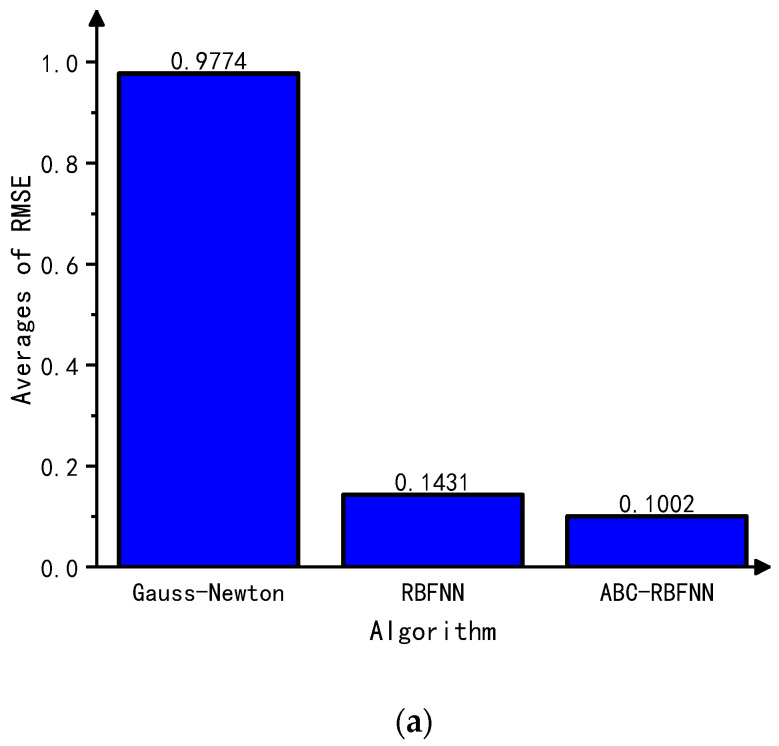
Average generalization test RMSE and ICC values. (**a**) Average RMSE values of the generalization test. (**b**) Average ICC values of the generalization test.

**Figure 11 sensors-23-07645-f011:**
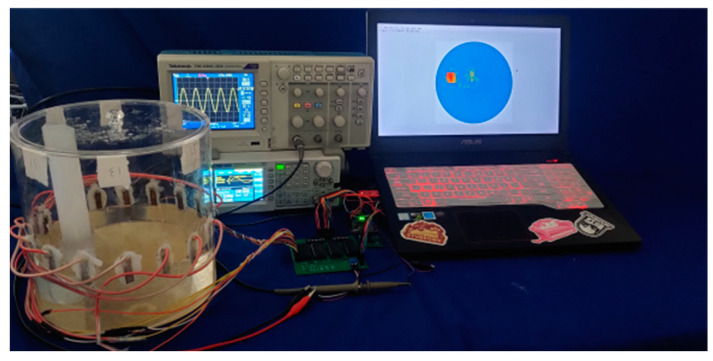
EIT experiment device.

**Figure 12 sensors-23-07645-f012:**
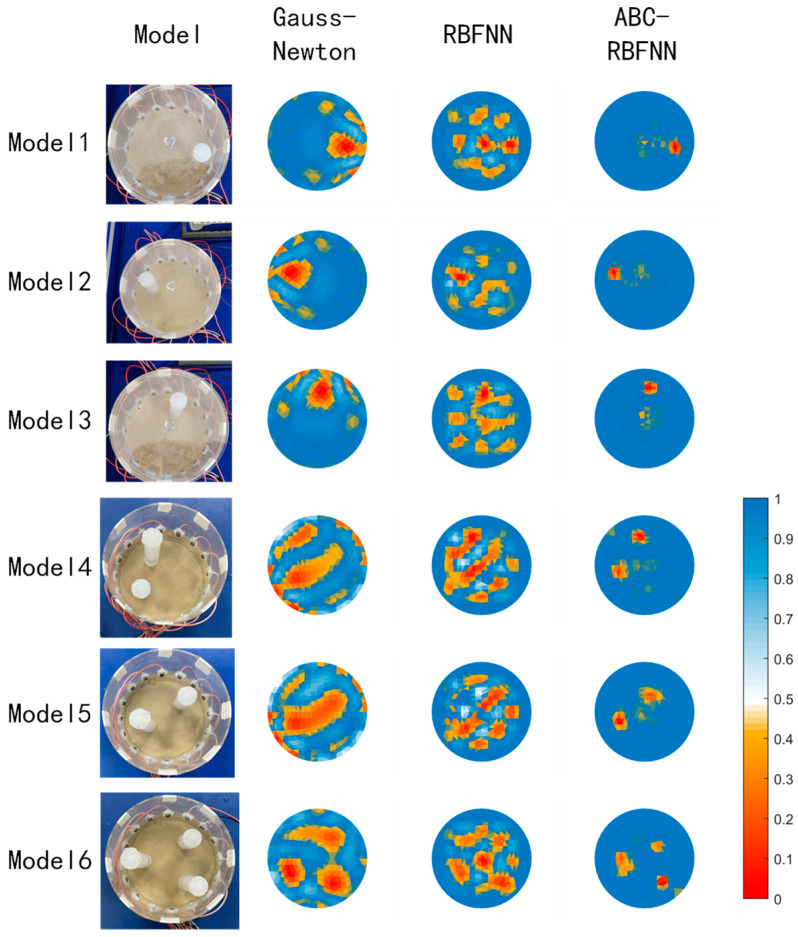
Imaging results of the experiment.

**Figure 13 sensors-23-07645-f013:**
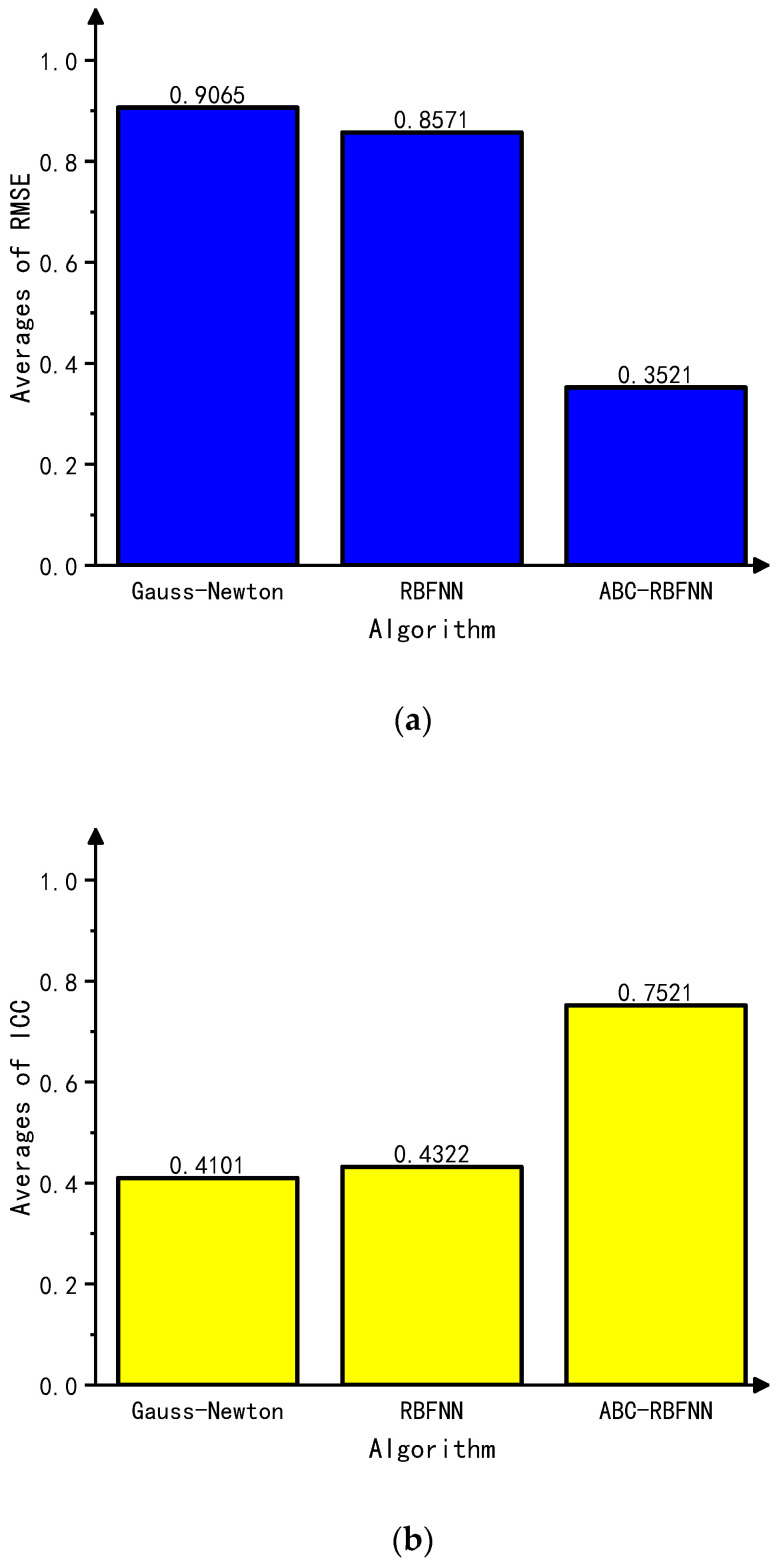
Average experimental RMSE and ICC values. (**a**) Average experimental of RMSE values. (**b**) Average experimental ICC values.

## Data Availability

Data is unavailable due to privacy restrictions.

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
