# Peer review of "Electrical Impedance Tomography of Industrial Two-Phase Flow Based on Radial Basis Function Neural Network Optimized by the Artificial Bee Colony Algorithm"

_sensors, 2023, doi:10.3390/s23177645_

Round 1
Reviewer 1 Report
This paper reported an EIT application using ABC algorithm to optimize the RBF neural network for two phase flow monitoring. Numerical and experimental results are shown to validate the proposed method. This is an interesting article and the paper is well written and the length of the article is justified by its contents however the manuscript is not suitable in present form and needs a major revision.
Please find the comments below
1. ABC algorithm described in the manuscript should include how it is modified for EIT. Also, more details of the algorithm complexity are required (i.e. computational burdens).
2. Authors have described they have used matlab for implementing training and testing of neural networks. It will be good to describe explicitly which platform is used for which task. Also in the manuscript there is no description of how the forward solution is obtained.
3. The conductivity values used in the numerical study have any practical value?
4. Authors have shown experimental studies using single target. Did the authors check for multiple targets detection using the proposed method?
5. What are the initial parameter values assigned in the ABC algorithm and how do they effect the final solution of proposed method. Also the information is missing about the RBF parameters for numerical and experimental studies.
6. Authors have introduced ABCRBFNN for two-phase flow imaging and the proposed method is tested with numerical test cases and experiments. The cases considered are mainly two phase situations. Is the proposed method applicable for multi-phase situations as well? Also, in the implementation of algorithm during training of NN is it assumed that the number of targets are known a priori.
7. The test cases for targets considered are mostly closed circular boundary. it would be interesting to see the results involving targets shape that are not used in training.
Minor editing of manuscript is required. Mainly manuscript should be checked for typo and spelling mistakes. Also, the equation notations have be defined and used in unified manner
Reviewer 2 Report
Dear authors and editors
This paper introduced the ABC-RBFNN method to the EIT imaging. In recent years, many different machine learning methods have been applied to inversion and imaging problem and these application has been published in many journals. The application of ABC-RBFNN method get a good imaging result than the traditional Gauss-Newton method. However, some details in this paper has to be improved. I recommend accept this paper after a moderate modification. The modification details are given below.
1. In the abstract section, “To address this issue, a new algorithm called Artificial Bee Colony-optimized Radial Basis Function Neural Network (ABC-RBFNN) is introduced”. Is the ABC-RBFNN a new algorithm you proposed? Or this is the first time the ABC-RBFNN is applied to this kind of imaging problem?
2. In the introduction section, I recommend to introduce more about the ABC method. I think much work has been made in this research field.
3. The size of the symbol in equation 1, 2 and 3 are not the same.
4. In the theory section, I suggest to explain why the ABC-RBFNN can get a good image result for this kind of problem.
5. What is the unit of the color bar in figure 3? Please give more detailed description of these models.
6. What is the title and unit of the y axis in figure 5, 7 and 9? The figures in this paper must be improved.
7. Are you using the same training set for gauss-newton and ABCRBFNN method? Could you please explain the characteristic of the training set? Are they completely random? If the training set has the characteristic of some red balls within the model, usually the Gauss-newton image can also get a good image result.
8. In the conclusion section, I recommend to enhance what is new in this paper.
9. The format of the referent should be modified. Reference 2 and 7 are in different format.
The Language is ok. Some momdification in details can make it better. However, It is also ok to be published in present form.
Round 2
Reviewer 1 Report
The authors have revised the manuscript however there are still a few minor issues before the acceptance of this paper.
Please find the comments below
1. Section 3 has to be revised. Especially section 3.3 which is the main idea of this paper needs to be explained clearly. Currently it is difficult to follow the steps.
2. Describe the working mechanism of the proposed method using a pseudo code or flow chart .
